# An Investigation of O-Demethyl Tramadol/Tramadol Ratio for Cytochrome P450 2D6 Phenotyping: The CYTRAM Study

**DOI:** 10.3390/pharmaceutics14102177

**Published:** 2022-10-12

**Authors:** Blandine De La Gastine, Soizic Percevault, Laurent Varin, Nicolas Richard, Fabienne Fobe, Benoît Plaud, Georges Daccache, Vincent Compere, Jean-Jacques Parienti, Antoine Coquerel, Magalie Loilier, Nathalie Bleyzac, Laurent Bourguignon, Sylvain Goutelle, Véronique Lelong-Boulouard

**Affiliations:** 1Hospices Civils de Lyon, GH Nord, Hôpital Pierre Garraud, 136 Rue du Commandant Charcot, 69005 Lyon, France; 2Hôpital Privé d’Antony, 1 Rue Velpeau, 92160 Antony, France; 3Service de Génétique, Centre Hospitalo-Universitaire Caen-Normandie, Avenue de la Côte de Nacre, 14000 Caen, France; 4UNICAEN, EA7450 BioTARGen, Normandie Université, 19 Rue Claude Bloch, 14000 Caen, France; 5Service d’Anesthésie-Réanimation, Centre Hospitalo-Universitaire Caen-Normandie, Avenue de la Côte de Nacre, 14000 Caen, France; 6Service d’anesthésie-réanimation-CTB, Université Paris Cité & AP-HP, Hôpital Saint-Louis, DMU PARABOL, 1 Avenue Claude Vellefaux, 75010 Paris, France; 7Service d’anesthésie-Réanimation, Centre Hospitalo-Universitaire Rouen-Normandie, 37 Boulevard Gambetta, 76000 Rouen, France; 8Direction de la Recherche Clinique et de l’innovation, Centre Hospitalo-Universitaire Caen-Normandie, Avenue de la Côte de Nacre, 14000 Caen, France; 9UNICAEN, INSERM U1075, COMETE, CYCERON, Normandie Université, 19 Rue Claude Bloch, 14000 Caen, France; 10Service de Pharmacologie, Centre Hospitalo-Universitaire Caen-Normandie, Avenue de la Côte de Nacre, 14000 Caen, France; 11Univ. Lyon, Université Lyon 1, UMR CNRS 5558, Laboratoire de Biométrie et Biologie Evolutive, 43 bd du 11 Novembre 1918, 69622 Villeurbanne, France; 12Univ. Lyon, Université Lyon 1, ISPB–Faculté de Pharmacie de Lyon, 8 Avenue Rockefeller, 69008 Lyon, France

**Keywords:** tramadol, *CYP2D6*, pharmacogenetics, pharmacogenomics

## Abstract

Cytochrome P450 2D6 (*CYP2D6*) gene polymorphisms influence the exposure to tramadol (T) and its pharmacologically active metabolite, O-demethyl tramadol (O-dT). Tramadol has been considered as a candidate probe drug for *CYP2D6* phenotyping. The objective of the CYTRAM study was to investigate the value of plasma O-dT/T ratio for *CYP2D6* phenotyping. European adult patients who received IV tramadol after surgery were included. *CYP2D6* genotyping was performed and subjects were classified as extensive (EM), intermediate (IM), poor (PM), or ultra-rapid (UM) *CYP2D6* metabolizers. Plasma concentrations of tramadol and O-dT were determined at 24 h and 48 h. The relationship between O-dT/T ratio and *CYP2D6* phenotype was examined in both a learning and a validation group. Genotype data were obtained in 301 patients, including 23 PM (8%), 117 IM (39%), 154 EM (51%), and 7 UM (2%). Tramadol trough concentrations at 24 h were available in 297 patients. Mean value of O-dT/T ratio was significantly lower in PM than in non-PM individuals (0.061 ± 0.031 versus 0.178 ± 0.09, *p* < 0.01). However, large overlap was observed in the distributions of O-dT/T ratio between groups. Statistical models based on O-dT/T ratio failed to identify *CYP2D6* phenotype with acceptable sensitivity and specificity. Those results suggest that tramadol is not an adequate probe drug for *CYP2D6* phenotyping.

## 1. Introduction

Cytochrome P450 2D6 (*CYP2D6*) is highly polymorphic with more than 100 variants described so far [1,2]. However, a few alleles are present in the vast majority of human genotypes. The most frequent alleles include *1 and *2 which are wild-type alleles with normal activity, *10 and *41 that have reduced but partial activity, and *3, *4, *5, *6, and *7 that are alleles with null activity [3]. Translation from genotype to phenotype depends on the activity of allele combinations. A recent article has reported a consensus recommendation on this topic [3]. Four metabolism phenotypes are usually considered: extensive metabolizers (EM) who show a “regular” *CYP2D6* activity, poor metabolizers (PM), intermediate metabolizers (IM), and ultrarapid metabolizers (UM). This latter phenotype is observed in individuals who carry multiple copies of wild-type alleles (e.g., *1/*1 × N, *2/*2 × N or *1/*2 × N). Those four groups represent approximately 50–70%, 5 to 10%, 10 to 30%, and 1 to 5% of the European population respectively [4,5,6]. The proportion of EM and IM vary widely across studies because the definition of allele activity and translation from genotype to the IM phenotype is not consistent [7].

*CYP2D6* participates in the metabolism of 20 to 25% of drugs [5]. Some of them have a narrow therapeutic window and mutations of *CYP2D6* may cause either overexposure or poor drug response [4]. Determining *CYP2D6* metabolizer status could be interesting to optimize response to sensitive *CYP2D6* substrate drugs. This can be done by genotyping or phenotyping. While *CYP2D6* genotyping has been widely developed in the last two decades [8], phenotyping based on a probe drug is still a relevant approach. Usually, this consists in measuring the concentrations of the parent drug and its metabolites of interest in blood and/or urine and comparing concentration ratios (parent/metabolite or vice-versa) to pre-defined cut-off values that discriminates genotype/phenotype groups.

Several probe drugs have been used for *CYP2D6* phenotyping, including debrisoquine, spartein and dextromethorphan, those being almost fully metabolized by *CYP2D6* [6]. However, debrisoquine and spartein are no longer available on the market and dextromethorphan is usually prescribed over only a few days as a cough suppressant. Administering such drugs for *CYP2D6* phenotyping raises ethical concerns in routine practice. Using a widely prescribed probe drug would facilitate *CYP2D6* phenotyping.

Tramadol is a weak opioid discovered in 1962 [9]. It is classified in the second step of the WHO analgesic ladder. Tramadol is widely prescribed for the treatment of chronic pain before requiring the third step, and in the management of acute post-operative pain. Mechanism of action of tramadol is mixed. It is an agonist of µ opioid receptors. It also increases serotonin and noradrenaline levels in the synapse by inhibiting the reuptake of these two transmitters at the presynaptic level [9,10,11]. *CYP2D6* is a significant metabolic pathway of tramadol. It is responsible for the conversion of tramadol to O-demethyl-tramadol (O-dT). The latter is an active metabolite with an approximately 300-fold higher affinity for µ-receptor than tramadol, and greater analgesic effect [11]. Both tramadol and O-dT have two stereogenic centers and exist as two main enantiomers. In patients who are administered tramadol to treat pain, the drug could be an interesting probe drug for *CYP2D6* phenotyping. In addition, *CYP2D6* phenotyping may be of interest to identify PM who may exhibit poor response to tramadol because of low exposure to the active metabolite O-dT [12]. In 2021, the Clinical Pharmacogenetics Implementation Consortium (CPIC) has published recommendations on tramadol therapy based on CYP2D6 phenotype, suggesting to avoid tramadol use in CYP2D6 PM subjects because of the risk of reduced analgesia [13].

Previous studies have investigated tramadol as a *CYP2D6* probe candidate. However, most of them used urinary concentrations and/or enantioselective assays which are not available in routine practice [14,15,16,17,18]. Levo et al. reported that the metabolic ratio (MR) tramadol/O-dT correlated with the number of active alleles of *CYP2D6* [19]. However, this was a *post-mortem* study whose results may not be predictive of in vivo results. Other in vivo studies confirmed the influence of *CYP2D6* genotype on the plasma concentrations of tramadol and O-dT, and on analgesic effect and tolerance of tramadol [9,20]. To our knowledge, no study has thoroughly investigated plasma O-dT/tramadol ratio as a method for *CYP2D6* phenotyping.

The primary objective of the CYTRAM study was to evaluate the predictive value of plasma O-dT/tramadol ratio for identifying *CYP2D6* PM individuals in patients who received tramadol for post-operative pain. The secondary objective was to investigate the overall value of plasma O-dT/tramadol ratio for *CYP2D6* phenotyping in this population.

## 2. Methods

### 2.1. Patients

CYTRAM was a prospective study conducted in four medical centers (three university hospitals and one private medical center). The study did not modify post-operative management and tramadol therapy, but it was considered as interventional because of *CYP2D6* genotyping. Inclusion criteria were as follows: age > 18 years, post-operative patient treated with intravenous tramadol for at least 24 to 48 h, European origin, and blood sampling at 24 h and 48 h in the post-operative monitoring. Exclusion criteria were as follows: patient who received opioid drugs before surgery, patient taking at least one *CYP2D6* inhibitor drugs before or during surgery, pregnancy or breast feeding, patients having one or more contraindications for taking tramadol in post-operative analgesia, and hepatocellular insufficiency (prothrombin time ratio < 70%). CYP inhibitors considered in the exclusion criterion were amiodarone, chloroquine, chlorpromazine, cimetidine, clomipramine, flecainide, fluoxetine, halofantrine, haloperidol, imatinib, levomepromazine, metoclopramide, moclobemide, paroxetine, promethazine, propafenone, quinidine, risperidone, and terbinafine.

Patients were administered IV tramadol for post-operative pain management. In two centers, 100 mg of IV tramadol was administered during surgery and 300 mg/24 h was subsequently administered by continuous IV infusion. In the third center, after a 100 mg initial IV dose during surgery, intermittent administration was done with 100 mg/6 h. In the fourth center, continuous infusion of tramadol with 300 mg/24 h was administered only after surgery. In all centers, the use of other analgesic drugs including morphine was possible whenever necessary for pain control.

Oral and written information was given, and each patient signed a written consent. The study was approved by an ethics committee (Comité de Protection des Personnes Nord Ouest) and the French national committee for data protection (Commission Nationale de l’Informatique et des Libertés). It was registered at the clinicaltrial.gov website (NCT00952159).

### 2.2. CYP2D6 Genotyping and Phenotyping

*CYP2D6* genotyping was performed on blood samples obtained at 24 h of tramadol therapy. Genotyping was centralized in the department of molecular genetics in Caen University hospital. A limited number of variants were investigated, *3, *4, *5, *6, *7, and *41 those being responsible for 98% of PM genotypes [3]. If none of those variants was identified, normal activity was assumed, with genotype denoted as *1–2/*1–2.

After DNA extraction, alleles *3, *4, *6, *7 and *41 were identified by TaqMan^®^ Drug Metabolism Genotyping Assays (Applied Biosystem). For whole gene duplications (*CYP2D6**nxn) and deletions (*CYP2D6**5), a long range PCR with specific duplication/deletion primers was used. Copy Number Variant (CNV) result was confirmed using a TaqMan^®^ Copy Number Assay.

Phenotypes were determined according to activity carried by each allele in accordance with guidelines [3,21]. The genotype to phenotype translation for the study population is shown in Table 1. We also defined as non-poor metabolizers (non-PM) all subjects who were not identified as PM. So, this group pooled individuals classified as IM, EM and UM (N = 278).

### 2.3. Determination of Tramadol and O-dT Concentrations

A single blood sample was obtained 24 h after surgery, just before the next planned dose and so corresponded to either plateau (continuous infusion) or trough concentration (intermittent administration). Both will be denoted as C_min_ for sake of simplicity. The O-dT/T ratio (ratio of concentrations in µg/L) at 24 h was calculated in each patient. In patients who were still under tramadol at 48h, C_min_ was also measured at this time 48 h and O-dT/T ratio calculated. Patients were excluded from the study if tramadol was stopped before 24 h of therapy.

All drug assays were centralized at the department of clinical pharmacology in Caen university hospital. Determination of tramadol (T) and O-dT concentrations in plasma was performed at 24 h by high performance liquid chromatography tandem mass spectrometry (LC-MSMS, Prominence Shimadzu system coupled with triple quadrupole API 4500 QTRAP ABSciex) with positive electrospray ionization after liquid/liquid extraction. Briefly, 200 µL of sample were fortified with 40 µL internal standards (tramadol-(13)C, 1 µg/L). Extraction was performed by liquid-liquid extraction using 200 µL sodium buffer (2N) and 2.5 mL ethyl acetate. After extraction, the chromatographic separation was achieved on biphenyl column using a mobile phase consisting of methanol and water both containing 1 mmol/L ammonium formiate and 0.02% formic acid. The column temperature was set at 40 °C and mobile phase flow rate at 0.5 mL/min in gradient mode with a total run time of 10 min. Calibration range includes 5 points (5, 10, 50, 300 and 500 µg/L). Mean precision for those 5 points was 50.4%, 77.1%, 104.2%, 102.3% and 99.1% for tramadol (r > 0.99) and 34.7%, 83%, 117.2%, 97.6%, and 100.7% for O-dT (r > 0.99). Three levels of internal controls are used at each range (20, 80 and 400 µg/L). The limit of quantification was 5 µg/L for tramadol and O-dT. The following transitions, using multiple reaction monitoring (MRM), were used for quantitation: 264/58 for tramadol, 250/58 for O-dT.

### 2.4. Statistical Analysis

The study sample size was based on the primary objective which was to identify *CYP2D6* PM individuals based on plasma O-dT/T ratio. We assumed a proportion of PM of 10% [19], as well as 90% sensitivity and 90% specificity of O-dT/T ratio in discriminating PM versus non-PM. Based on those assumptions, a sample size of 300 was necessary to estimate sensitivity and specificity with a precision of 11% and 2%, respectively. The original estimated enrollment was fixed at N = 320 to account for data exclusion.

We first compared mean values of O-dT/T ratio between phenotype groups by using the Student T test. The Student T test, the Welsh test or the Mann Whitney test were used for comparisons of other variables between phenotype groups in accordance with sample size and other application criteria. Statistical significance was set at *p* ≤ 0.05. In addition, the histograms and empirical distributions of O-dT/T ratio values were plotted and visually compared between groups.

Several approaches were used to identify cut-off values of O-dT/T ratio for *CYP2D6* phenotype discrimination. First, an empirical search without any model was performed by testing various cut-off value and computing sensibility and specificity for the discrimination between PM and non-PM subjects in the entire dataset.

Then, more sophisticated and robust statistical approaches were used to identify variables predicting *CYP2D6* phenotype. We randomly split the overall dataset into a learning (80%) and validation (20%) dataset. Random sampling was set to respect the proportions of phenotypes from the original full dataset. The validation group of 50 patients had phenotype proportions as follows: 25 EM (51%), 20 IM (39%), 4 PM (8%) and 1 UM (1%). Then, we used univariate and multivariate classification trees (CT), which are machine learning methods, to identify cut-off values of the metabolic ratio that discriminated *CYP2D6* phenotypes in the learning dataset. Univariate models only included O-dT/T ratio at 24 h or 48 h. Then, a bivariate model including the ratio at both times was assessed. Multivariate models including all available variables were also investigated (metabolic ratios at 24 h and 48 h, sex, age, weight, serum creatinine, prothrombin time). The Gini diversity index was used for defining binary splits in trees. In order to avoid overfitting, a minimal number of five observations was set in child nodes. Predictive performance of the prediction trees was assessed by computing sensitivity (Se), specificity (Sp), positive (PPV) and negative (NPV) predictive values in the learning dataset. Finally, the robustness of the prediction trees was assessed by computing their predictive performance in the validation dataset.

Statistical analysis was performed by using the Orange toolbox [22] and the Statistica software (version 13.3, Tibco, Palo Alto, CA, USA). Plots were builtwith R.

## 3. Results

### 3.1. Patients’ Characteristics

Patients were enrolled from September 2009 to June 2011. A total of 401 European patients were eligible during this time frame. The study flow-chart is summarized in Figure 1. *CYP2D6* genotyping was performed in 310 patients. Nine patients were excluded from the study after genotype testing for various reasons as shown in Figure 1. Genotype data were further analyzed in 301 patients. Based on the alleles identified, the distribution of *CYP2D6* phenotype was as follows: 23 PM (8%), 117 IM (39%), 154 EM (51%), and 7 UM (2%). Regarding tramadol concentration at 24 h, data from 297 out of 301 patients were available. Data from 4 patients were excluded (two EM and two PM individuals): one because of missing blood sample at 24 h, and three because of outlier results. Concentration data at 48 h were available in 227 patients. Patients excluded from the analysis at 48 h included those in whom tramadol was stopped between 24 and 48 h, as well as three outliers. The four outliers had tramadol and/or O-dT concentrations that deviated from the mean value of more than 10 standard deviations. We assumed that an error occurred in blood sampling, such as sampling during the tramadol IV infusion or insufficient rinse before sampling.

The characteristics of the study population are presented in Table 2. Tramadol doses were not statistically different between *CYP2D6* phenotype groups (*p* = 0.815 at 24 h and *p* = 0.504 at 48 h).

Tramadol concentrations at 24 h and 48 h were not significantly different between *CYP2D6* phenotype groups. By contrast, O-dT concentrations were significantly lower in PM than in non-PM subjects. In addition, PM showed significantly lower values of O-dT/T ratio at 24 h and 48 h compared with non-PM, EM and IM individuals (*p* < 0.01). The difference was also significant for PM + IM versus EM + UM (*p* < 0.05). Of note, metabolic ratio in UM subjects was not significantly greater than that of EM (*p* = 0.694).

Large overlap was observed between *CYP2D6* phenotypes in the distribution of O-dT/T ratios, as shown in Figure 2 and Figure 3.

### 3.2. Prediction of CYP2D6 Phenotype Based on O-dT/T Ratio

In the model-free analysis of the entire dataset, a metabolic ratio cut-off ≤0.1 was associated with the best performance in discriminating PM from non-PM, with Se = 87%, Sp = 94.5%, PPV = 87%, and NPV = 94.5% at 24 h. Results for the ratio calculated at 48 h were as follows: Se = 87.5%, Sp = 83.8%, PPV = 88%, and NVP = 83%. However, model-based analysis with external validation showed poorer results. Univariate and bivariate classification tree models-based O-dT/T ratio at 24 and 48 h showed low to moderate predictive performance in both the learning and validation datasets, as shown in Table 3 and Table 4. No model achieved the predefined acceptable Se and Sp of 90% in the learning set and performance worsened in the validation group. Multivariate classification tree model did not improve predictive performance (data not shown).

## 4. Discussion

This is one of the largest studies that evaluated tramadol as a probe drug for *CYP2D6* phenotyping. To our knowledge, this is the first study that investigated tramadol as a probe drug for *CYP2D6* phenotyping in real life setting with robust validation in a test group.

Our results confirm the association between *CYP2D6* genotype/phenotype and exposure to the active metabolite of tramadol, O-dT. Patients with PM status showed the lowest mean value of O-dT plasma concentrations and O-dT/T ratio at 24 h and 48 h compared with non-PM subjects. In addition, the mean value of O-dT/T ratio increased with the number of active *CYP2D6* alleles in PM, IM and EM groups. Those results are consistent with previous publications regarding the influence of *CYP2D6* gene polymorphisms and phenotype on exposure to tramadol and its active metabolite [15,16,17,18,20]. UM patients did not show an increased value of the metabolic ratio compared with EM, but the number of UM subjects in the study (N = 7) preclude from a definitive conclusion. Of note, the conversion of tramadol to O-dT by *CYP2D6* also has implications for drug-drug interactions [23]. When O-dT/T ratio was used to predict *CYP2D6* phenotype, a cut-off value of 0.1 appeared to separate PM and non-PM with good predictive performance in the entire dataset. Unfortunately, performance worsened when prediction was applied in a validation group after data splitting. Indeed, no single value of the metabolic ratio could discriminate the various phenotypes with acceptable performance. This is consistent with the large overlap of distributions of the metabolic ratio between phenotypes (see Figure 2 and Figure 3).

Other studies have reported similar results. Pedersen et al. could not identify the expected bimodal distribution of urinary MR (+)-Tramadol/(+)O-dT in subjects who received 50 mg of tramadol PO [15]. Paar et al., found the same result for the urinary MR (0 to 24 h) of tramadol/O-dT after oral administration of 50 mg of tramadol [14]. Indeed, even if *CYP2D6* contributes to tramadol metabolism into O-dT and influences the drug pharmacokinetics (PK), its contribution to overall tramadol body clearance, approximately 30% [24], is probably not sufficient for use as a probe drug, at least in the conditions of our study. It has been shown that other metabolic pathways are involved the metabolism of tramadol and its metabolites. Another important metabolic pathway of tramadol is N-demethylation, that involves CYP3A4 and CYP2B6. In addition, both N-demethyl tramadol and O-demethyl tramadol undergo further metabolism, with participation of some phase II enzymes [9,25,26]. Each of these pathways has its own inherent variability contributing to overall variability of tramadol and metabolite concentrations. This explains the overlap in the distributions of the metabolic ratio between phenotypes observed in our study.

Based on our study results, trough concentration of tramadol and O-dT cannot be considered as robust predictors of *CYP2D6* phenotype. However, measuring such concentrations may still be useful to investigate insufficient pain relief or unexpected toxicity in individual patients.

This study has several limitations. Only the major *CYP2D6* loss-of-function alleles were search by genotyping. Phenotypes and their proportions might have been different based on a larger variant search. The metabolic ratio O-dT/T was based on single trough sampling at 24 h and 48 h. This sparse sampling design was chosen to simplify phenotyping in routine clinical practice for the study and for potential future applications of the approach in real life setting. However, such sparse sampling is likely to influence the observed PK variability in plasma concentrations and concentration ratios. Other sampling times, as well as richer PK data could lead to different results. This is an area for further research. As mentioned above, other CYP pathways contribute to tramadol and metabolite clearance, including *CYP3A4* and *CYP2B6* [9]. Gene polymorphisms of those pathway were not investigated at the time of this study. With the genomics tools currently available (e.g., whole exome sequencing or whole genome sequencing), a more exhaustive genotyping could be investigated in future studies. A drug with a *CYP2D6*-mediated clearance higher than that of tramadol may be a better probe for *CYP2D6* phenotyping. However, the probe drug should also have other characteristics such as good tolerance for phenotyping. For example, yohimbine, which is metabolized at approximately 90% by *CYP2D6* has been investigated as a probe drug. However, it has many adverse effects and is rarely administered in clinical practice because of its poor tolerance [27]. The main alternative to phenotyping based on probe drugs is direct genotyping of *CYP2D6*. Techniques, as well as availability and cost if genotyping assay are improving [8,28]. In addition, the development of recommendation for standardizing the translation from genotype to phenotype should make interpretation of genotype easier [3,13].

## 5. Conclusions

In this study, *CYP2D6* genotype/phenotype influenced exposure to the active metabolite of tramadol (O-dT) and concentration ratios (O-dT/T) in patients who received IV tramadol after surgery. However, O-dT/T ratio based on tramadol trough concentration did not provide robust prediction of *CYP2D6* PM phenotype. No single value of the metabolic ratio and no multivariate model could discriminate the various phenotypes with acceptable performance in a test group. Although a study with richer pharmacokinetic data would be necessary to confirm findings, this study suggests that tramadol is not an adequate probe drug for *CYP2D6* phenotyping.

## Figures and Tables

**Figure 1 pharmaceutics-14-02177-f001:**
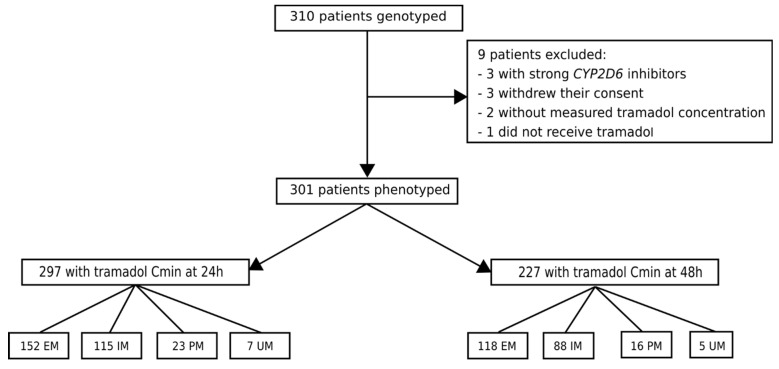
Flowchart of data analysis.

**Figure 2 pharmaceutics-14-02177-f002:**
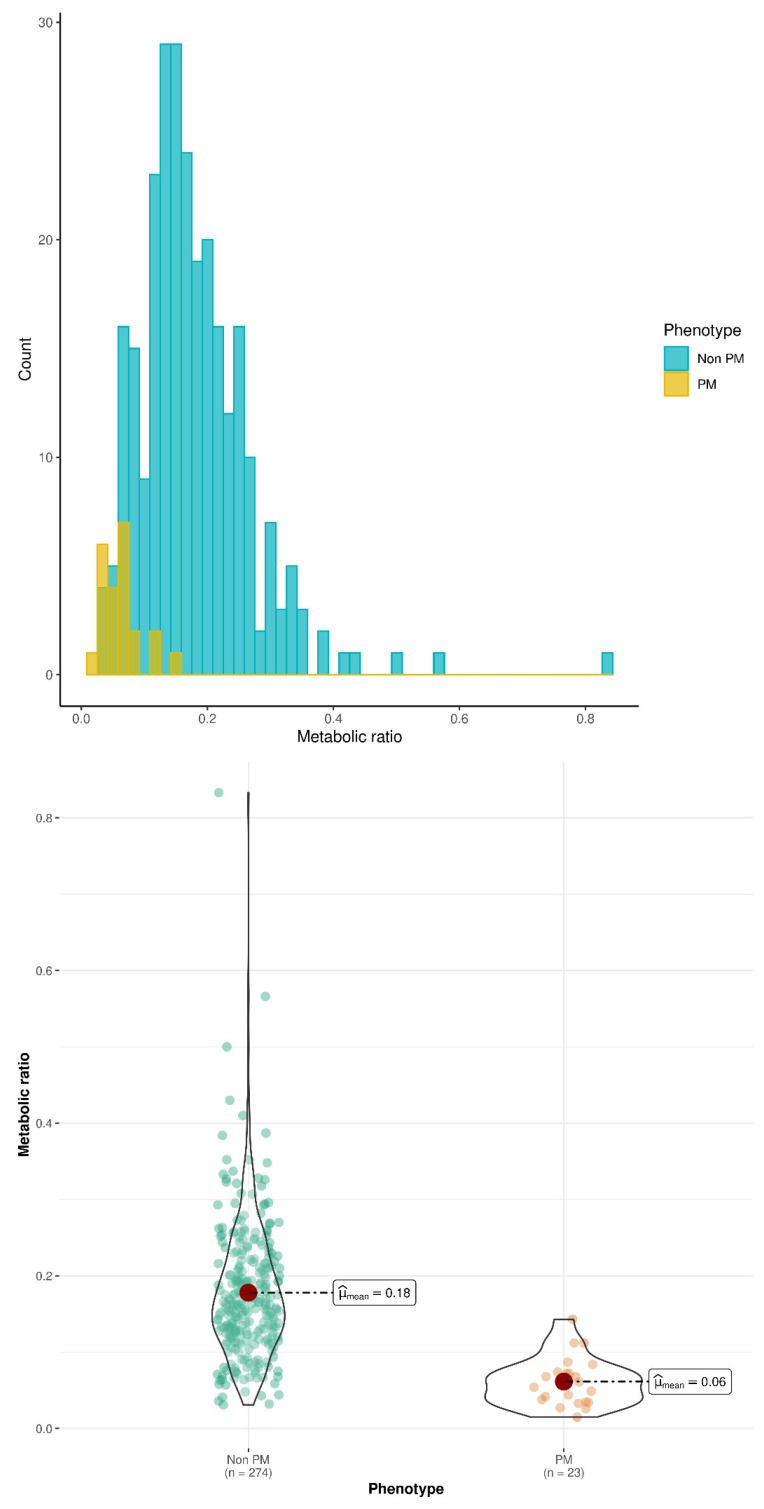
Distribution of O-dT/T ratio at 24 h in PM and non-PM. Upper panel, histogram of phenotype counts; lower panel, violin plot of metabolic (O-dT/T) ratios. The colored dots represent individual data, the large red filled circle represents the mean value, the black solid line shows the probability density of data.

**Figure 3 pharmaceutics-14-02177-f003:**
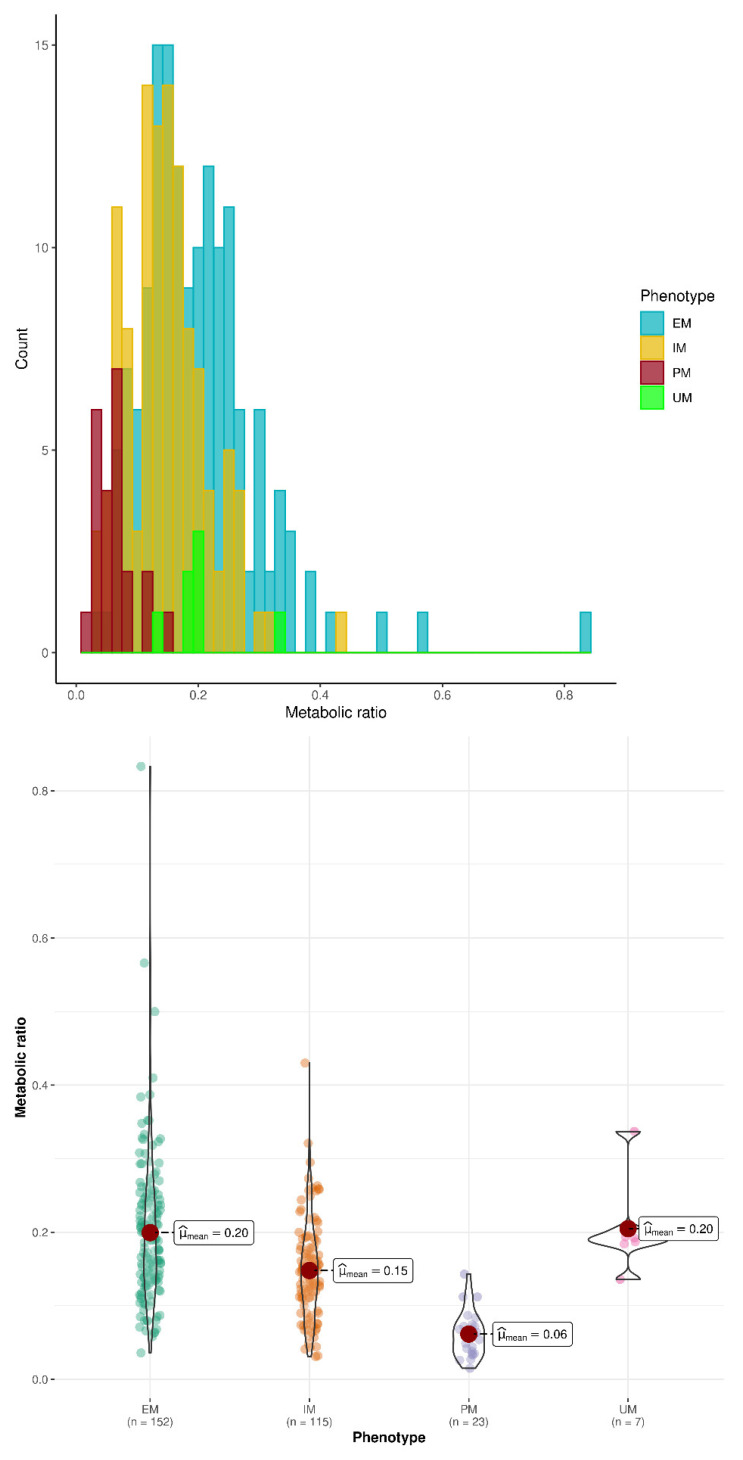
Distribution of O-dT/T ratio at 24 h in all phenotype groups. Upper panel, histogram of phenotype counts; lower panel, violin plot of metabolic (O-dT/T) ratios. The colored dots represent individual data, the large red filled circle represents the mean value, the black solid line shows the probability density of data.

**Table 1 pharmaceutics-14-02177-t001:** CYP2D6 genotypes and phenotypes in the study population.

Extensive Metabolizers. (*n* = 154)	Poor Metabolizers (*n* = 23)	Intermediate Metabolizers (*n* = 117)	Ultra-Rapid Metabolizers (*n* = 7)
*1–2/*1–2 (*n* = 112) *1/*41 (*n* = 42)	*3/*3 (*n* = 1) *4/*4 (*n* = 13) *3/*4 (*n* = 6) *4/*5 (*n* = 3)	*1–2/*3 (*n* = 5) *1–2/*4 (*n* = 96) *1/*5 (*n* = 13) *1/*6 (*n*= 1) *1/*7 (*n* = 2)	*1–2/*1–2 × N (*n* = 7)

**Table 2 pharmaceutics-14-02177-t002:** Patients’ characteristics according to phenotype.

	PM	Non-PM	IM	EM	UM	PM-IM	EM-UM	All Subjects
Number of subjects (number with concentration data at 24 h)	23 (23)	278 (274)	117 (115)	154 (152)	7 (7)	140 (138)	161 (159)	301 (297)
Sex (% male)	57	55	60	53	71	48	52	56
Age	60.5 (12.7)	59.7 (14.5)	59.5 (13.8)	60.0 (14.9)	59.65 (16.4)	59.7 (13.7)	60.0 (17.8)	59.9 (14.3)
Digestive surgery	77.3%	61.2%	55.5%	64.9%	71.4%	59.1%	65.2%	62.6%
Orthopedic surgery	22.7%	26.1%	30.9%	23.2%	14.3%	29.6%	22.8%	25.5%
Urologic surgery	0%	11.2%	11.8%	11.3%	0%	9.9%	10.8%	10.5%
Other surgery	0%	1.5%	1.8%	0.7%	14.3%	1.5%	1.3%	1.4%
Weight (kg)	74.9 (14.3)	77.6 (16.9)	76.9 (17.0)	78.4 (17.1)	73.3 (8.7)	75.9 (16.4)	78.4 (16.9)	77.3 (16.7)
Serum creatinine (µmol/L)	77.7 (27.0)	79.4 (22.2)	80.8 (23.4)	78.7 (21.4)	73.1 (17.2)	80.4 (24.3)	78.3 (21.2)	79.2 (22.7)
PT (%)	97.1 (6.2)	96.5 (7.1)	95.4 (8.2)	97.2 (6.3)	99.3 (1.8)	95.7 (7.9)	97.3 (6.2)	96.6 (7.1)
Mean dose of tramadol at 24 h (mg)	433 (83)	430 (78)	428 (73)	430 (83)	457 (50)	429 (75)	431 (82)	430 (79)
Mean dose of tramadol at 48 h	353 (61)	340 (54)	336 (54)	342 (54)	357 (50)	338 (55)	343 (54)	341 (54)
Tramadol concentration (µg/L) at 24 h	487 (215)	452 (216)	452 (213)	449 (219)	495 (213)	458 (214)	451 (219)	454 (217)
Tramadol concentration (µg/L) at 48 h	545 (263)	420 (263)	389 (268)	442 (262)	406 (119)	414 (278)	441 (258)	429 (267)
O-dT concentration (µg/L) at 24 h	31 ab (22)	76 a (45)	67 bc (46)	81 c (43)	103 (56)	61 d (45)	82 d (44)	72 (46)
O-dT concentration (µg/L) at 48 h	33 ab (24)	68 a (50)	60 b (54)	74 (47)	71 (32)	55 (52)	74 (46)	66 (50)
O-dT/T at 24h Mean (SD)	0.061 ab (0.031)	0.178 a (0.090)	0.145 bc (0.066)	0.198 c (0.099)	0.204 (0.058)	0.134 d (0.070)	0.200 d (0.098)	0.172 (0.092)
O-dT/T at 48 h Mean (SD)	0.063 ab (0.032)	0.182 a (0.109)	2.407 b (0.121)	0.189 (0.099)	0.174 (0.053)	0.154 (0.119)	0.191 d (0.098)	0.177 (0.110)
O-dT/T at 24 h Min-Max Median	0.015–0.143 0.061	0.031–0.833 0.163	0.031–0.43 0.146	0.036–0.833 0.189	0.136–0.337 0.192	0.015–0.43 0.129	0.036–0.833 0.143	0.015–0.833 0.157
O-dT/T at 48 h Min-Max Median	0.025–0.044 0.064	0.039–1 0.166	0.044–1 0.154	0.039–0.8 0.167	0.116–0.268 0.165	0.025–1 0.186	0.039–0.8 0.170	0.025–1 0.158

^a^: *p* < 0.01 between PM and Non-PM; ^b^: *p* < 0.01 between PM and IM; ^c^: *p* < 0.05 between IM and EM; ^d^: *p* < 0.05 between PM-IM and EM-UM. Abbreviations: EM, extensive metabolizers; IM, intermediate metabolizers; non-PM, non-poor metabolizers; O-dT, O-demethyl-tramadol; PM, poor metabolizers, PT = Prothrombine Time; T, tramadol; UM, ultra-rapid metabolizers. Data are presented as mean (SD) unless otherwise stated.

**Table 3 pharmaceutics-14-02177-t003:** Cut-off values of O-dT/T ratio at 24 h and 48 h and predictive performance of the univariate classification tree model for phenotype prediction.

	PM vs. Non-PM	PM vs. IM	IM vs. EM	IM vs. EM + UM	EM vs. UM	UM vs. Non-UM	PM-IM vs. EM-UM
	O-dT/T ratio at 24 h						
O-dT/T ratio cut-off	≤0.0745	≤0.0745	≤0.196	≤0.196	≤0.182	≤0.184	≤0.132
Sensitivity	50% (84%)	50% (84%)	65% (86%)	65% (86%)	0% (100%)	100% (17%)	42% (55%)
Specificity	93% (90%)	85% (86%)	52% (45%)	50% (44%)	60% (17%)	47% (34%)	77% (80%)
PPV	40% (42%)	40% (55%)	52% (54%)	50% (53%)	0% (5%)	4% (1%)	63% (70%)
NPV	96% (100%)	89% (96%)	65% (81%)	65% (82%)	94% (100%)	100% (94%)	59% (68%)
	O-dT/T ratio at 48 h						
O-dT/T ratio cut-off	≤0.076	≤0.098	≤0.087	≤0.098	≤0.270	≤0.114	≤0.087
Sensitivity	75% (75%)	75% (92%)	20% (20%)	20% (20%)	100% (20%)	0% (0%)	29% (30%)
Specificity	89% (93%)	75% (78%)	92% (96%)	92% (96%)	12% (50%)	67% (73%)	92% (96%)
PPV	43% (81%)	38% (58%)	67% (77%)	67% (77%)	4% (2%)	0% (0%)	78% (86%)
NPV	98% (98%)	94% (98%)	59% (62%)	60% (63%)	100% (93%)	97% (97%)	59% (62%)

Each cell shows two results, for the validation dataset and (learning dataset), respectively. Abbreviations: EM, extensive metabolizers; IM, intermediate metabolizers; non-PM, non-poor metabolizers; non-UM, non-ultrarapid metabolizers; NPV, Negative Predictive Value; O-dT, O-demethyl-tramadol; PM, poor metabolizers; PPV, Positive Predictive Value; PT = Prothrombine Time; T, tramadol; UM, ultra-rapid metabolizers.

**Table 4 pharmaceutics-14-02177-t004:** Cut-off values of O-dT/T ratio at 24 h and 48 h and predictive performance of the bivariate classification tree model for phenotype prediction.

	PM vs. Non-PM	PM vs. IM	IM vs. EM + UM	IM vs. EM	EM vs. UM	UM vs. Non-UM	PM-IM vs. EM-UM
O-dT/T ratio cut-off at 24 h	≤0.0745	≤0.089	≤0.196	≤0.196	≤0.195	≤0.183	≤0.172
O-dT/T ratio cut-off at 48 h	NA	NA	≤0.087	≤0.087	NA	NA	≤0.087
Sensitivity	75% (83%)	75% (89%)	20% (20%)	20% (20%)	0% (83%)	0% (0%)	29% (29%)
Specificity	91% (90%)	75% (81%)	92% (96%)	92% (96%)	40% (50%)	33% (94%)	92% (96%)
PPV	43% (38%)	38% (49%)	67% (76%)	67% (76%)	0% (7%)	0% (37%)	78% (85%)
NPV	98% (99%)	94% (97%)	60% (63%)	59% (69%)	88% (98%)	94% (94%)	59% (62%)

Each cell shows two results, for the validation dataset and (learning dataset), respectively. Abbreviations: EM, extensive metabolizers; IM, intermediate metabolizers; non-PM, non-poor metabolizers; non-UM, non-ultrarapid metabolizers; NA, not applicable; NPV, Negative Predictive Value; O-dT, O-demethyl-tramadol; PM, poor metabolizers; PPV, Positive Predictive Value; PT = Prothrombine Time; T, tramadol; UM, ultra-rapid metabolizers.

## Data Availability

Data are available upon reasonable request from the corresponding author.

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
