# Peer review of "An Investigation of O-Demethyl Tramadol/Tramadol Ratio for Cytochrome P450 2D6 Phenotyping: The CYTRAM Study"

_pharmaceutics, 2022, doi:10.3390/pharmaceutics14102177_

Round 1

Reviewer 1 Report

1)     General: The authors should eliminate data where they cannot phenotype the individuals? Non PM for non-poor metabolizers seem to be in that group.

2)     General: The authors should clearly correlate all the groups.

3)     Line 175 and other lines: Cmin should be subscripted.

4)     Line 190: Tramadol calibration should be provided.

5)     Figure 1: Top of the flowchart is cut off.

6)     Table 2: Is Non PM for non-poor metabolizers? What does that mean? The authors should clearly explain: PM-IM and EM-UM.

7)     Table 2: The row with PM for poor metabolizer, etc. was that determined by genotyping and/or phenotyping?

8)     Figure 2: The gaussian fits are very poor to the data. The authors should be able to get a better fit. The authors should represent the fraction of individuals as points and fit the gaussians to it. If the gaussian fits and the bars represent two different datasets, they should be on separate graphs.

9)     Figure 2: I am not sure what Non-Poor Metabolizer means. I guess extensive metabolizer and ultra metabolizer. The authors should genotype those people.

10)  Figure 3: Horrendous fit of gaussians to the data. Their gaussian distributions do not make any sense. The authors should represent the fraction of individuals as points and fit the gaussians to it. If the gaussian fits and the bars represent two different datasets, they should be on separate graphs.

11)  The authors need to explain the software that they used to analyze their data.

12)  Fonts in the figures should be larger.

Author Response

1/ General: The authors should eliminate data where they cannot phenotype the individuals? Non PM for non-poor metabolizers seem to be in that group.

  1. We considered as non-PM all subjects who were not classified as PM (Table 1). So, this group included IM, EM and UM and the genotype/phenotype of those individuals was known. We agree that this definition was not clear. This has been clarified in the methods section.

2/ General: The authors should clearly correlate all the groups.

  1. We are afraid, but this comment is unclear to us. What does this mean?

3/ Line 175 and other lines: Cmin should be subscripted.

  1. Done.

4/ Line 190: Tramadol calibration should be provided.

  1. Details about calibration of tramadol and o-demethyl tramadol assays have been added.

5/ Figure 1: Top of the flowchart is cut off.

  1. Yes, this was cut in the text file. The top is not cut-off in the original figure file. This could be handled with the production staff if the manuscript is accepted.

6/ Table 2: Is Non PM for non-poor metabolizers? What does that mean? The authors should clearly explain: PM-IM and EM-UM.

  1. Yes, this means non-poor metabolizers. The definition has been added in the method section, as explained above, and the abbreviation explained in the table footnote.

7/ Table 2: The row with PM for poor metabolizer, etc. was that determined by genotyping and/or phenotyping?

  1. Of course, the phenotype indicated in Table 2 are consistent with the definition presented in Table 1.

8/ Figure 2: The gaussian fits are very poor to the data. The authors should be able to get a better fit. The authors should represent the fraction of individuals as points and fit the gaussians to it. If the gaussian fits and the bars represent two different datasets, they should be on separate graphs.

  1. In Figure 2 and 3, the fitted Gaussian curves were just overlaid with the observation counts. Indeed, the graph scales were not consistent between the curves and the counts. So, they should not be interpreted in terms of goodness-of-fit.

Our aim was just to show the overlap between distributions. However, fitting a Gaussian distribution was not appropriate, because of data sparsity in some phenotype groups (e.g., UM).

New figures are provided in the revised manuscript: new histograms without Gaussian curves and with larger font as well as violin plots showing the overlap between distributions.

9)     Figure 2: I am not sure what Non-Poor Metabolizer means. I guess extensive metabolizer and ultra metabolizer. The authors should genotype those people.

  1. This comment has been addressed (see comments #1 and #6). Actually, those people were genotyped.

10)  Figure 3: Horrendous fit of gaussians to the data. Their gaussian distributions do not make any sense. The authors should represent the fraction of individuals as points and fit the gaussians to it. If the gaussian fits and the bars represent two different datasets, they should be on separate graphs.

  1. This comment has been addressed (see comment #8)

11/ The authors need to explain the software that they used to analyze their data.

  1. Statistical analysis was performed by using Orange toolbox and Statistica software. Plots were made with R. These details have been added in the method section.

12/ Fonts in the figures should be larger.

  1. The figures have been redrawn with a larger font.

Reviewer 2 Report

The manuscript is well written and clear. Tramadol was adminw don'tistered intravenously.

What are the author's expectations had the drug been given orally? The data in Table 1

are not consistent with the distributions of metabolized phenotypes listed in introduction especially for 

both EMs and IMs. How do authors explain this apparent inconsistency?

Page 3,line 81 has a misselling

Author Response

The manuscript is well written and clear. Tramadol was adminw don'tistered intravenously.

What are the author's expectations had the drug been given orally?

  1. We believe that conclusions would hold true for tramadol administered by oral route, because the route of administration does not influence the drug elimination pathways. Our results are consistent with those from studies with oral tramadol. For example, Parr et al. (MS reference # 13), could not clearly discriminate the distributions of the apparent log of the tramadol O-de-methylation ratio between PM and IM, in contrast with the distributions of the metabolic ratio of sparteine, a known CYP2D6 probe drug.

The data in Table 1 are not consistent with the distributions of metabolized phenotypes listed in introduction especially for both EMs and IMs. How do authors explain this apparent inconsistency?

  1. Thank you for pointing this discrepancy. It is clear that the proportion of IM in our study (40%) was greater than that mentioned in the introduction. Indeed, the phenotype frequencies that we indicated in the introduction was based on quite old studies, mostly performed in Northern America. In those older studies, a limited number of loss-of-function alleles were investigated. Because CYP2D6*1 and *2 are the most common default assignments, it is likely that the proportion of those alleles and that of the EM phenotype were overestimated. Also, genotypes including *1 or *2 and a loss-of-function allele have been inconsistently classified as IM or EM in the literature. More recent data support a higher proportion of alleles with reduced activity and IM phenotype especially in Europe (see https://doi.org/10.1038/gim.2016.80. The sentence in the introduction has been revised, and this reference has been added.

Page 3,line 81 has a misselling

  1. Correction done